# Life-Course Associations between Blood Pressure-Related Polygenic Risk Scores and Hypertension in the Bogalusa Heart Study

**DOI:** 10.3390/genes13081473

**Published:** 2022-08-18

**Authors:** Xiao Sun, Yang Pan, Ruiyuan Zhang, Ileana De Anda-Duran, Zhijie Huang, Changwei Li, Mengyao Shi, Alexander C. Razavi, Lydia A. Bazzano, Jiang He, Tamar Sofer, Tanika N. Kelly

**Affiliations:** 1Department of Epidemiology, Tulane University School of Public Health and Tropical Medicine, New Orleans, LA 70112, USA; 2Department of Medicine, College of Medicine, University of Illinois at Chicago, Chicago, IL 60612, USA; 3Jiangsu Key Laboratory of Preventive and Translational Medicine for Geriatric Diseases, Department of Epidemiology, School of Public Health, Medical College of Soochow University, Suzhou 215123, China; 4Department of Internal Medicine, Emory University, Atlanta, GA 30322, USA; 5Department of Biostatistics, Department of Medicine, Harvard University, Boston, MA 02115, USA

**Keywords:** hypertension, polygenic risk score, genetic predisposition, blood pressure, childhood risk factor, genetics

## Abstract

Genetic information may help to identify individuals at increased risk for hypertension in early life, prior to the manifestation of elevated blood pressure (BP) values. We examined 369 Black and 832 White Bogalusa Heart Study (BHS) participants recruited in childhood and followed for approximately 37 years. The multi-ancestry genome-wide polygenic risk scores (PRSs) for systolic BP (SBP), diastolic BP (DBP), and hypertension were tested for an association with incident hypertension and stage 2 hypertension using Cox proportional hazards models. Race-stratified analyses were adjusted for baseline age, age2, sex, body mass index, genetic principal components, and BP. In Black participants, each standard deviation increase in SBP and DBP PRS conferred a 38% (*p* = 0.009) and 22% (*p* = 0.02) increased risk of hypertension and a 74% (*p* < 0.001) and 50% (*p* < 0.001) increased risk of stage 2 hypertension, respectively, while no association was observed with the hypertension PRSs. In Whites, each standard deviation increase in SBP, DBP, and hypertension PRS conferred a 24% (*p* < 0.05), 29% (*p* = 0.01), and 25% (*p* < 0.001) increased risk of hypertension, and a 27% (*p* = 0.08), 29% (0.01), and 42% (*p* < 0.001) increased risk of stage 2 hypertension, respectively. The addition of BP PRSs to the covariable-only models generally improved the C-statistics (*p* < 0.05). Multi-ancestry BP PRSs demonstrate the utility of genomic information in the early life prediction of hypertension.

## 1. Introduction

Hypertension is a common condition that affects nearly one-third of the world’s adult population. As a leading risk factor for cardiovascular disease and death [1,2,3], high blood pressure (BP) is a substantial contributor to global morbidity and mortality despite the availability of effective therapeutic strategies for its treatment and control. Hence, the early diagnosis and prevention of high BP are critical for reducing hypertension’s attributable disease burden [4]. Over the past few decades, extensive studies have been conducted to characterize clinical and lifestyle risk factors for hypertension [5,6,7,8]. Early-life risk factors, including childhood BP, have been reported to predict hypertension in adulthood [9,10,11]. It is well-documented that genetic factors also play an important role in determining BP-related traits [12,13]. Genome-wide association studies (GWAS), which leverage genomics information from millions of participants, have become the primary tool for deciphering the genetic basis of hypertension and have all together identified more than 1,000 BP loci [14,15,16,17,18,19,20,21,22,23]. Genome-wide polygenic risk scores (PRSs), which leverage GWAS summary statistics and can combine thousands or millions of risk variants, have recently gained popularity for examining the cumulative effects of genetic variants on disease susceptibility [19,24,25,26,27,28]. While prospective studies have noted strong associations between BP PRSs and incident hypertension [25,29,30,31,32], only a handful of studies have assessed the relevance of a genome-wide PRS beyond clinically measured BP [24,30,33,34], and all have been conducted in populations of European or Asian ancestry. Since BP can be easily and inexpensively assessed in a clinic, data on the added value of genetics are needed to discern the clinical utility of genomic information for risk stratification purposes. 

Similar to the effects of lifestyle and behaviors, genetic effects accumulate gradually over time, with previous studies demonstrating stronger correlations between PRSs and measured phenotypes with an increasing mean population age [35,36]. These observations suggest that genetic data may be particularly useful for disease prediction in early life, capturing important information that cannot yet be measured clinically. In this study, we hypothesized that genetic predisposition to high BP could help identify individuals in childhood who might be at higher risk for hypertension. We tested this hypothesis by leveraging recently validated multi-ancestry genome-wide PRSs for systolic BP (SBP), diastolic BP (DBP), and HTN [37]. We then examined the associations between these BP PRSs and hypertension development and midlife BP among a sample of two races of 1201 Bogalusa Heart Study (BHS) participants with up to 16 carefully collected measures of BP from childhood to midlife. We compared the performance of the BP PRS in predicting hypertension and high BP across the lifespan before and after adjusting for the measured childhood BP.

## 2. Materials and Methods

### 2.1. Study Participants

The BHS is a population-based long-term study examining the natural history of cardiovascular disease (CVD) and risk factors from childhood to adulthood among participants of two races (35% Black and 65% White) from Bogalusa, Louisiana. From 1973 until today, 9 surveys have been conducted among children and adolescents aged 4 to 17 years, and 11 surveys have been conducted among adults aged 18 to 51 years who were examined previously as children. Detailed descriptions of the BHS design and methods have been reported previously [38]. The current study included a total of 1201 participants from the BHS who were examined at least once in childhood and once in adulthood and had existing genome-wide genetic data. 

### 2.2. Genotyping, PRS Derivation, and Validation

Genome-wide genotyping was performed on the Illumina Human610 BeadChip using methods described previously [39,40,41,42]. In brief, stringent quality control removed SNPs with a low call rate or a poor cluster separation score. Untyped genotypes were imputed from the TOPMed reference panel using the TOPMed Imputation Server developed by the University of Michigan [43,44,45]. Variants with an imputation quality of <0.3 were further removed. Genomic ancestry principal components (PCs) were calculated using the GCTA software tool version v1.26.0 [46] among the Black and White participants, separately.

The current study leveraged previously constructed and validated multi-ancestry PRSs for SBP, DBP, and HTN [37]. In brief, the SBP and DBP PRSs were derived using summary statistics from BP GWAS of up to 318,891 participants of the Million Veteran Program (MVP) [20]. The HTN PRS was calculated as a sum of the SBP and DBP PRS above and derived using summary statistics from HTN GWAS of up to 453,010 “pan ancestry” UK Biobank participants (https://pan.ukbb.broadinstitute.org)(Accessed on 16 April 2022). Clump-and-threshold methods, implemented in PRSice-2 software version v2.3.3 [47], were used to construct all PRSs. Totals of 24,807, 149,559, and 234,228 variants were included for the SBP, DBP, and HTN PRS construction, respectively, among the BHS participants. For each BHS participant, the PRS was calculated by multiplying each variant’s effect size by its dosage and summing dosage-by-effect size products across all variants included in the PRS. The PRSs were examined continuously and in quartile categories within each race group.

### 2.3. Phenotype Measurements

Stringent protocols were utilized to collect clinical data on the BHS participants [48,49]. Anthropometric measures were obtained with participants in lightweight clothing without shoes. At each visit, the body weight and height were measured twice to the nearest 0.1 kg and 0.1 cm, respectively. The body mass index (BMI) was calculated by dividing the mean body weight in kilograms by the mean height in meters squared. In childhood, the blood pressure was measured in duplicate from the right arm using a mercury sphygmomanometer while the participants were in a relaxed, sitting position [50]. In adulthood, the blood pressure was measured in triplicate on the right arm of participants using the Omron HEM 907XL digital blood pressure device after 5 minutes in the sitting position. For 12 hours prior to the study visit, participants were advised to avoid eating, smoking, intake of caffeine and alcohol, and physical activity. Fifteen mmHg and 10 mmHg were added to the raw SBP and DBP values, respectively, if a participant was taking antihypertensive medication [51]. The mean values of the SBP and DBP from each study visit were used for analyses. For individuals aged one to thirteen years, hypertension was defined as SBP or DBP ≥95th percentile (based on age, sex, and height percentiles), SBP ≥ 130 mmHg, DBP ≥ 80 mmHg, or use of antihypertensive medications [52]. For participants older than 13, hypertension was defined as SBP ≥ 130 mmHg, DBP ≥ 80 mmHg, or use of antihypertensive medications, and stage 2 hypertension was defined as SBP ≥ 140 mmHg or DBP ≥ 90 mmHg, or use of antihypertensive medications [53].

### 2.4. Statistical Analyses

The characteristics of the study participants were calculated and stratified by race in childhood (baseline) and midlife (most recently completed visit). Continuous variables are presented as the median and interquartile range (IQR) or the means and standard deviations (SDs), and categorical variables are presented as numbers and percentages (%). The differences in the normally and non-normally distributed continuous variables between the two race groups were evaluated using a t-test and the Wilcoxon rank sum test, respectively. The differences between the categorical variables were evaluated using the chi-square test.

#### 2.4.1. Association between the PRSs and Incidence of Hypertension and Stage 2 Hypertension 

Cox proportional hazards models were used to assess the association between the continuous and categorized (quartiles) PRSs and the incidence of hypertension and stage 2 hypertension. Participants with hypertension at baseline were excluded from the analysis, and follow-up years was used as the time scale. The null model contained only study covariables: baseline age, baseline age square, sex, baseline BMI, and ancestry principal components (PCs). Ten ancestry PCs were adjusted in the Black participants and 3 were adjusted in the White participants to appropriately account for genetic diversity [54]. Model 1 included childhood BP and adjusted for covariables in the null model. Model 2 included the PRSs and adjusted for covariables in the null model. Model 3 included the PRSs and childhood BP and adjusted for covariables in the null model. Specifically, the childhood SBP was used in Models 1 and 3 when testing the SBP PRS, the childhood DBP was used when testing the DBP PRS, and the childhood mean arterial pressure (MAP) was used when testing the HTN PRS. The MAP was calculated as the DBP plus one-third of pulse pressure (the difference between the SBP and DBP). Harrell’s C-statistics were calculated to measure the goodness-of-fit of the models. To evaluate the added value of the PRSs and measured childhood BP to the models, Models 1 and 2 were compared to the null model, and Model 3 was compared to Models 1 and 2 using paired concordance tests [55,56]. For PRS quartiles, the *p* values for the linear trend were also assessed. 

#### 2.4.2. Association between the PRSs and Midlife Blood Pressure

The associations between the continuous and categorized (quartiles) PRSs and the last visit blood pressure (SBP and DBP) were assessed using multivariable linear regression. As described above, the null model included baseline age, baseline age square, sex, baseline BMI, and ancestry PCs. Model 1 included childhood BP and adjusted for covariables in the null model. Model 2 included the PRS and adjusted for covariables in the null model. Model 3 was adjusted for covariables in the null model along with the PRS and childhood BP. To evaluate the improvement in variation explained by adding childhood BP or PRSs, the adjusted R-square values were calculated for each model. Models 1 and 2 were both compared with the null model and Model 3 was compared with both Models 2 and 1 using F-tests. For the PRS quartiles, the P values for the linear trend were also assessed. 

All analyses were performed using R version 4.0.3 and SAS 9.4. 

## 3. Results

Table 1 presents the characteristics of the BHS participants in childhood and adulthood according to race. The White participants were more likely to be male and have smaller PRSs than the Black participants. The follow-up time between the two race groups was similar. In childhood, the mean age, mean BMI, mean SBP, mean DBP, and percent with hypertension were similar between the Black and White race groups. In the most recent adulthood visit, the Black participants had higher BMI, SBP, DBP, and percent of hypertension and stage 2 hypertension compared to the White adults. Thirty-nine percent of the Black adult participants were taking antihypertensive medication, which is a percentage significantly higher than that of the White participants (23.8%).

### 3.1. Association between the PRSs and Incidence of Hypertension and Stage 2 Hypertension

The associations between the continuous SBP, DBP, and HTN PRSs with the incident hypertension, as well as their corresponding C-statistics, are presented in Table 2. Among the Black participants, in the model adjusting for the covariables and the PRS (Model 2), each SD increase in the SBP PRS conferred a 42% (*p* = 4.87 × 10^−3^) higher risk of hypertension and a 75% (*p* =1.22 × 10^−4^) higher risk of stage 2 hypertension. Each SD increment in the DBP PRS conferred a 24% (*p* = 8.15 × 10^−3^) and a 52% (*p* = 2.16 × 10^−5^) higher risk of hypertension and stage 2 hypertension, respectively. After adjusting for childhood BP, the associations remained significant, with each SD increase in SBP PRS associated with a 38% higher risk of hypertension and a 74% higher risk of stage 2 hypertension (*p* = 9.62 × 10^−3^ and 1.59 × 10^−4^), and each SD increase in DBP PRS associated with a 22% higher risk of hypertension and a 50% higher risk of stage 2 hypertension (*p* = 1.69 × 10^−2^ and 3.34 × 10^−5^, respectively). The associations between the HTN PRS with hypertension and stage 2 hypertension were not significant in any of the models assessed. Compared to the null model, including childhood BP or the continuous PRSs significantly improved the model fit (*p* < 0.05). Adding the childhood BP or DBP PRS always improved the model fit for the two outcomes, while adding the SBP PRS to the model that was already adjusted for childhood SBP improved the model for the stage 2 hypertension outcome (*p* = 8.03 × 10^−4^). 

Among the White participants, in the model unadjusted for the baseline BP (Model 2), each SD increment increase in the PRS for SBP, DBP, and HTN conferred a 29% (*p* = 1.6 × 10^−2^), 30% (*p* = 8.53 × 10^−3^), and 27% (*p* = 1.02 × 10^−4^) higher risk of hypertension, respectively. Likewise, each SD increment increase in the PRS for SBP, DBP, and HTN conferred a 33% (*p* = 3.84 × 10^−2^), 42% (P = 8.57 × 10^−3^), and 44% (*p* = 1.13 × 10^−5^) higher risk of stage 2 hypertension, respectively. In the model further including the baseline BP (Model 3), each SD increment increase in the PRS for SBP, DBP, and HTN conferred a 24% (*p* = 4.49 × 10^−2^), 29% (*p* = 1.23 × 10^−2^), and 25% (*p* = 2.54 × 10^−4^) higher risk of hypertension, respectively. Each SD increment increase in the DBP PRS and HTN PRS conferred a 1.39 (*p* = 1.31 × 10^−2^) and 1.42 (*p* = 2.29 × 10^−5^)-fold higher risk of stage 2 hypertension, respectively, while the SBP PRS was not significantly associated with hypertension.

The findings of the quartile PRS analyses were generally consistent with those of the continuous PRS analyses, identifying the dose-response increases in hypertension risks before and after adjusting for childhood BP (Figure 1). For example, among the Black BHS participants, increasing the SBP PRS quartile was associated with stage 2 hypertension before and after adjusting for childhood BP (P for linear trend (P_linear_= 4.67 × 10^−3^ and 5.95 × 10^−3^, respectively), with the last PRS quartile conferring a respective 2.01-fold and 2.05-fold increased risk of stage 2 hypertension compared to the first quartile. Likewise, compared to the first quartile, the last quartile of the DBP PRS conferred a 1.58-fold and 1.52-fold higher risk of hypertension (P_linear_= 1.22 × 10^−2^ and 2.03 × 10^−2^, respectively) and a 2.34-fold and 2.29-fold higher risk of stage 2 hypertension (P_linear_= 1.00 × 10^−4^ and 1.39×10^−4^, respectively) before and after adjusting for childhood BP, respectively. Among the White participants, the dose-response associations between the SBP PRS quartile and hypertension and the DBP PRS quartile and stage 2 hypertension were observed before adjusting for childhood BP (P_linear_= 3.11 × 10^−2^ and 4.86 × 10^−2^, respectively); the last PRS quartiles conferred a respective 1.22-fold increased risk of hypertension and a 1.38-fold increased risk of stage 2 hypertension, respectively. Similar but non-significant trends were observed after adjusting for childhood BP (P_linear_= 0.05 and 0.07, respectively). The HTN PRS quartile was significantly associated with hypertension before and after adjusting for childhood MAP (P_linear_= 1.01 × 10^−4^ and 1.85 × 10^−4^, respectively), with the last quartile conferring a 1.60-fold and 1.58-fold higher risk of hypertension, respectively, compared to the first quartile. Likewise, the HTN PRS quartile was also associated with stage 2 hypertension before and after adjusting for childhood BP (P_linear_= 9.22 × 10^−5^ and 1.57 × 10^−4^, respectively), with the last quartile conferring a respective 1.95-fold and 1.90-fold higher risk of stage 2 hypertension compared to the first quartile.

### 3.2. Association between the PRSs and Midlife Blood Pressure

As shown in Table 3, significant or marginal associations between the SBP and DBP PRSs and the midlife BP values were observed among the Black BHS participants, both before and after adjusting for childhood BP. For example, before and after adjusting for childhood SBP, the Black participants demonstrated 5.8 mmHg and 4.2 mmHg higher midlife SBP values, respectively, per SD increase in SBP PRS (*p* = 0.03 and 0.05, respectively). Likewise, the midlife DBP values were, on average, 2.5 mmHg and 2.3 mmHg higher per SD increase in DBP PRS before and after adjusting for childhood DBP, respectively (*p* = 7.45 × 10^−3^ and 0.01, respectively). No associations between the HTN PRSs and BP were observed in the Black participants. In the Whites, all the BP PRSs were significantly associated with the midlife BP both before and after adjusting for childhood BP. For example, before and after adjusting for childhood SBP, the White participants demonstrated 4.5 mmHg and 3.9 mmHg higher midlife SBP values, respectively, per SD increase in SBP PRS (*p* = 6.42 × 10^−4^ and 2.42 × 10^−3^, respectively). Likewise, the midlife DBP values were, on average, 2.7 mmHg and 2.6 mmHg higher per SD increase in DBP PRS before and after adjusting for childhood DBP, respectively (*p* = 1.37 × 10^−3^ and 2.01 × 10^−3^, respectively). In general, and in both the Black and White BHS participants, the explained variance significantly improved when adding the BP PRSs to the models, even among the models including the childhood BP measures. 

The findings of the PRS quartile analyses were generally consistent with those of the continuous PRS analyses (Figure 2). For example, the Black participants in the highest DBP PRS quartile had 5.62 mmHg and 5.08 mmHg higher DBPs, respectively, at midlife compared to those in the lowest PRS quartile (P_linear_=1.14 × 10^−2^ and 2.14 × 10^−2^, respectively) before and after adjusting for childhood DBP. A similar but non-significant trend was observed for the SBP PRS, and no association was observed for the HTN PRS. In the White participants, significant associations were observed across all three PRSs and midlife BP both before and after adjusting for childhood BP. For example, those in the highest quartile of the SBP PRS had 4.31 mmHg and 3.83 mmHg higher midlife SBP before and after adjusting for childhood SBP, respectively (P_linear_=1.25 × 10^−3^ and 3.03 × 10^−3^, respectively), compared to those in the lowest quartile. Similarly, those in the highest DBP PRS quartile had 2.48 mmHg and 2.34 mmHg higher DBPs, respectively, at midlife compared to those in the lowest PRS quartile (P_linear_=1.21 × 10^−2^ and 1.43 × 10^−2^, respectively) before and after adjusting for childhood DBP. 

## 4. Discussion

In the first study to examine associations between genome-wide BP PRSs and the development of hypertension from childhood to midlife, we identified strong associations that were independent of the measured childhood BP levels. For example, the White BHS participants in the highest quartile of the HTN PRS had a significant 1.6-fold increased risk of hypertension compared to those in the first quartile, even after adjusting for childhood BP. As expected, the associations between the BP PRSs were generally stronger in magnitude when assessing more severe hypertension, with, for example, the highest quartile of the HTN PRS conferring a 1.9-fold increased risk of stage 2 hypertension in Whites when compared to the first quartile. Consistent with the associations observed for hypertension, higher BP PRSs were generally associated with higher midlife BPs across both race groups, both before and after adjusting for childhood BP measures. In our evaluation of predictive information, we observed significantly improved C-statistics and adjusted r-square values when adding the BP PRSs to the models that included measured childhood BP, highlighting the potential clinical utility of genomic information for risk stratification purposes in early life. When comparing our multi-ancestry BP PRS associations across the race groups, the SBP and DBP PRSs performed as well, if not better, in the Black compared to the White participants. As an example, each SD increase in the SBP PRS conferred a significant 1.7-fold increased risk of stage 2 hypertension in Blacks compared to the significant but relatively attenuated 1.2-fold increased risk of stage 2 hypertension observed in Whites. In contrast, the HTN PRS was only associated with BP endpoints in the White participants. Overall, our findings could have important public health and clinical significance and suggest that genomic information could help to facilitate targeted primordial hypertension prevention strategies in diverse populations with the identification of individuals who have a genetic susceptibility to high BP.

Our study identified independent associations between BP PRSs and incident hypertension and midlife BP in the BHS cohort. While this is the only study to have assessed associations between genome-wide PRSs and hypertension independently from measured BP, several studies have investigated this relationship using PRSs comprising a small number of GWAS-identified variants, with similar findings [24,32,34,57,58]. The Young Finns study is the only other study to have assessed the independent association between a BP PRS and the development of hypertension starting in childhood, identifying a significant 1.3-fold increase in odds of adult hypertension for every standard deviation increase in the BP PRS [57]. Although their PRS was limited to only 29 variants, it is interesting to note that the effect size observed in this Finnish population is consistent with the effect sizes that we observed for hypertension in the White participants using our various genome-wide BP PRSs. However, the previous study used a higher DBP cut point for defining hypertension (DBP≥85 mmHg). When using this same threshold in post-hoc analyses, the hazard ratios in Whites ranged from 1.41 to 1.42, suggesting that our genome-wide PRSs had modestly improved performance compared to the restricted PRS used previously. In aggregate, our study adds to the accumulating data showing that genetic information is independently associated with the life-course development of BP and hypertension. Furthermore, we provide some of the first evidence that BP PRSs are valuable for risk prediction purposes in diverse populations, providing further impetus for the expansion of genomics research to under-represented racial and ethnic groups.

In addition to identifying independent associations between BP PRS and hypertension and midlife BP, our study further demonstrates the ability of BP PRSs to improve the prediction of these endpoints. A handful of studies on this topic have been conducted previously [28,34,57,58,59]. Similar to our work, the Young Finns study examined the added value of a BP PRS to a multivariable model that included childhood SBP. Although their BP PRS was restricted to only 29 variants, this PRS improved hypertension prediction based on C-statistics [57]. In the only study to have evaluated a genome-wide BP PRS for the prediction of hypertension beyond measured BP at baseline, Vaura and colleagues found a significant 0.7% improvement in the C-statistic when adding the BP PRS to a model that included measured BP [28]. These findings are similar to the results of the current study, which identified significant C-statistic improvement values ranging from 0.4% to 1.5%. Overall, our data highlight the potential clinical utility of genetic information in identifying individuals in early life who might be at an increased risk of developing hypertension for targeted prevention strategies.

Using a multi-ancestry PRS, our study is among the first to demonstrate strong, independent associations between a BP PRS and incident hypertension in a Black sample. Surprisingly, associations between the SBP and DBP PRSs and hypertension incidence appeared to be larger in magnitude in the Black compared to the White BHS participants. This finding was unanticipated because the PRSs were derived from summary statistics from a BP GWAS meta-analysis of MVP participants, which comprised a 69% non-Hispanic White and 19% non-Hispanic Black sample. Historically, PRSs have demonstrated decreased performance in non-White populations because most GWASs, which are used to construct the PRS, are conducted in ancestrally European samples [60]. Hence, differences in allele frequencies, linkage disequilibrium, and genetic landscapes (i.e., ancestry-specific genetic variations and differences due to selective pressures) between European and other ancestral groups likely attenuate PRS signals in non-European populations [61,62,63,64,65,66]. Given the higher frequency of White participants in the MVP, we hypothesized that the PRS would perform somewhat better in this population. Although our finding was unexpected, a previous study assessing the performance of the BP PRS in Black and White CARDIA participants showed that the Black participants in the high PRS percentiles developed hypertension earlier than the White participants in higher PRS percentiles when compared to their counterparts in lower PRS percentiles [37]. Given the limited number of studies with this type of data, it is difficult to determine whether this might be a statistical anomaly or whether genetic susceptibility could be more deterministic in Black participants. Either way, future research in this area is needed. In contrast, the HTN PRS performed better in the White compared to the Black participants. Although this PRS was derived from “pan-ancestry” GWAS summary statistics, the derivation cohorts had a substantially higher frequency of ancestrally European participants (89%) compared to the GWAS used to derive the SBP and DBP PRSs.

Our study has several strengths. The BHS includes longitudinally and carefully collected data on cardiometabolic risk factors from childhood. With an average follow-up time of 37 years, the BHS offered a unique opportunity to assess the associations between PRSs and BP-related traits independent from the influence of childhood BP. Furthermore, because this study was conducted in a sample of two races of participants, we were able to evaluate the performance of the PRS in non-White participants, which have been underrepresented in genomics research. The PRSs used in this study were derived from the summary statistics of a large, multi-ancestry GWAS that were tested among 10,314 diverse participants using a novel approach and evaluated in 40,201 individuals [37]. Representing some of the first BP PRSs generated in diverse samples, our study demonstrates strong associations in non-White participants. Despite these important strengths, we acknowledge certain limitations. The BP measurement methods in the BHS changed over the long follow-up period. However, all study participants had BP measures obtained in childhood and, more recently, in adulthood. Therefore, any systematic differences related to the measurement methods should apply relatively equally to the study participants and have minimal impact on the overall study findings. The associations between the BP PRSs and midlife BP were relatively imprecise in the Black BHS participants, making it difficult to compare associations across the race groups. This is likely related to the relatively smaller sample size of this subgroup and the higher frequency of antihypertensive medication use, making it difficult to accurately estimate true BP values in these participants. In addition, the HTN PRS appeared to perform poorly in the Black participants, which is likely related to the ancestral composition of the derivation GWAS. In contrast, this PRS displayed a somewhat enhanced performance in Whites, suggesting that the HTN PRS may provide superior information to that of the SBP or DBP PRS. Hence, HTN PRSs derived from larger and more diverse multi-ancestry GWASs are needed. 

## 5. Conclusions

In conclusion, we provide important evidence of the utility of multi-ancestry BP PRSs for the early-life prediction of incident hypertension and midlife BP among both White and Black individuals. The observed associations are independent of childhood BP, suggesting that genetic data can capture information on BP that may not be clinically measurable in early life. We also demonstrated that a multi-ancestry BP PRS performed well in both Black and White BHS participants, highlighting the importance of expanded genomics research and the development of PRSs in diverse samples. These data add to the accumulating evidence that genomics information could help to identify high-risk subgroups for lifestyle and behavioral interventions. The finding that genetic information may be useful in childhood offers a unique opportunity for primordial prevention strategies, possibly enabling targeted early-life interventions, such as a low-sodium diet or increased physical activity, prior to a clinical manifestation of high BP.

## Figures and Tables

**Figure 1 genes-13-01473-f001:**
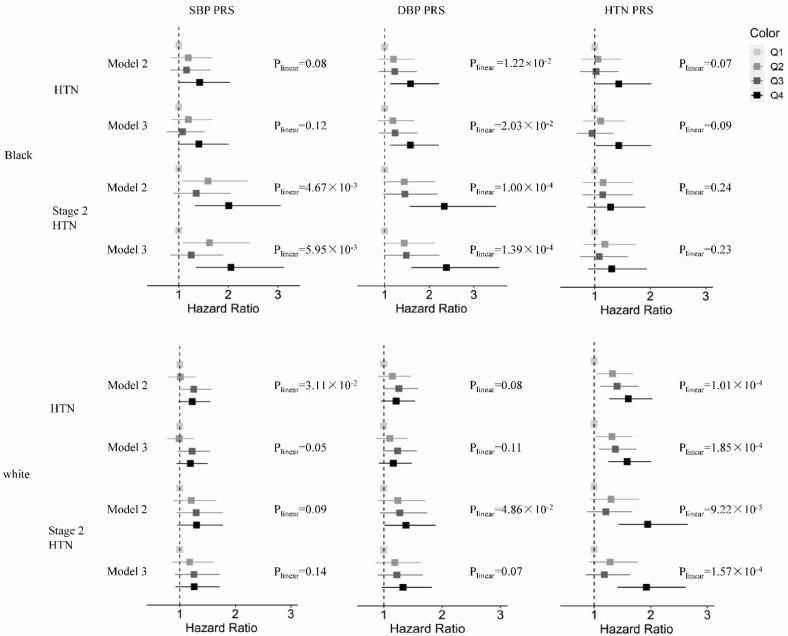
The association between PRS quartiles by race and the incidence of hypertension and stage 2 hypertension before (Model 2) and after (Model 3) adjusting for childhood BP.

**Figure 2 genes-13-01473-f002:**
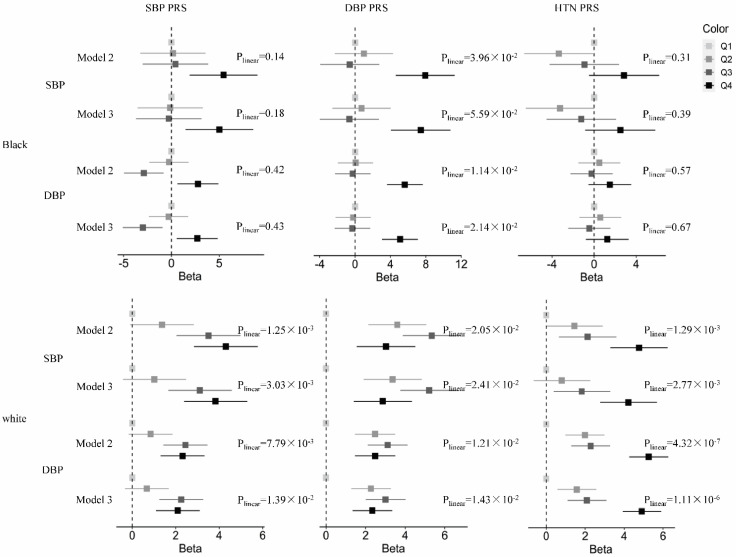
The association between PRS quartiles by race and midlife BP before (Model 2) and after (Model 3) adjusting for childhood BP.

**Table 1 genes-13-01473-t001:** Characteristics of Bogalusa Heart Study participants (N = 1201).

	Blacks (N = 369)	Whites (N = 832)	*p*
Male, n (%)	143 (38.8)	375 (45.1)	0.04
Time, yrs, median (IQR)	36.2 (9.99)	37.1 (8.88)	0.55
SBP PRS, mean (SD)	−4.90 × 10^−3^ (6.10 × 10^−4^)	−7.00 × 10^−3^ (4.60 × 10^−4^)	<0.0001
DBP PRS, mean (SD)	−8.00 × 10^−4^ (2.20 × 10^−4^)	−1.30 × 10^−3^ (1.20 × 10^−4^)	<0.0001
HTN PRS, mean (SD)	1.43 × 10^−5^ (1.59 × 10^−5^)	−1.86 × 10^−5^ (1.45 × 10^−5^)	<0.0001
Childhood *			
Age, yrs, mean (SD)	9.56 (2.98)	9.94 (3.28)	0.05
BMI, kg/m^2^, mean (SD)	17.6 (3.75)	17.8 (3.47)	0.34
SBP, mmHg, mean (SD)	99.1 (10.69)	100 (9.83)	0.08
DBP, mmHg, mean (SD)	61.88 (8.72)	62.03 (8.17)	0.78
Hypertension, n (%)	16 (4.34)	46 (5.55)	0.38
Stage-2 hypertension (%)	2 (0.54)	0 (0.00)	0.03
Antihypertension medication, n (%)	0 (0.00)	0 (0.00)	NA
Midlife ^†^			
Age, yrs, mean (SD)	44.6 (7.38)	45.4 (6.99)	0.07
BMI, kg/m^2^, mean (SD)	32.8 (9.15)	30.2 (6.93)	<0.0001
SBP, mmHg, mean (SD)	126 (19.7)	118 (14.0)	<0.0001
DBP, mmHg, mean (SD)	82.6 (11.8)	78.2 (9.7)	<0.0001
Hypertension, n (%)	265 (71.8)	462 (55.6)	<0.0001
Stage-2 hypertension (%)	197 (53.4)	286 (34.4)	<0.0001
Antihypertension medication, n (%)	144 (39.0)	198 (23.8)	<0.0001

* Baseline study visit. ^†^ Most recently completed study visit. SBP: Systolic blood pressure; PRS: Polygenic risk score; DBP: Diastolic blood pressure; HTN: Hypertension; NA: Not available; BMI: Body mass index.

**Table 2 genes-13-01473-t002:** Association between PRSs and childhood BP with hypertension outcomes.

	Model 1 ^§^				Model 2 ^†^				Model 3 ^‡^				
BP PRS	Hazard Ratio (CI)	*p*-Value	Harrell’s C	*p*-Value (M1 vs. Null *)	Hazard Ratio (CI)	*p*-Value	Harrell’s C	*p*-value (M2 vs. Null *)	Hazard Ratio (CI)	*p*-Value	Harrell’s C	*p*-Value (M3 vs. M2)	*p*-Value (M3 vs. M1)
Black													
Hypertension													
SBP PRS													
PRS	-	-	0.629	1.07 × 10^−3^	1.42 (1.11, 1.82)	4.87 × 10^−3^	0.615	1.99 × 10^−2^	1.38 (1.08, 1.77)	9.62 × 10^−3^	0.631	2.87 × 10^−3^	0.07
Measured SBP	1.26 (1.07, 1.48)	5.71 × 10^−3^			-	-			1.23 (1.05, 1.45)	1.14 × 10^−2^			
DBP PRS													
PRS	-	-	0.633	4.21 × 10^−7^	1.24 (1.06, 1.46)	8.15 × 10^−3^	0.615	8.20 × 10^−3^	1.22 (1.04, 1.43)	1.66 × 10^−2^	0.634	2.13 × 10^−6^	2.49 × 10^−2^
Measured DBP	1.29 (1.11, 1.49)	6.15 × 10^−4^			-	-			1.27 (1.1, 1.46)	1.22 × 10^−3^			
HTN PRS													
PRS	-	-	0.638	1.27 × 10^−7^	1.18 (0.99, 1.40)	0.06	0.620	0.06	1.14 (0.96, 1.36)	0.13	0.643	3.29 × 10^−7^	0.15
Measured MAP	1.34 (1.14, 1.56)	2.67 × 10^−4^			-	-			1.32 (1.13, 1.55)	4.90 × 10^−4^			
Stage 2 Hypertension													
SBP PRS													
PRS	-	-	0.613	1.55 × 10^−2^	1.75 (1.31, 2.32)	1.22 × 10^−4^	0.618	5.00 × 10^−4^	1.74 (1.30, 2.31)	1.59 × 10^−4^	0.627	2.21 × 10^−2^	8.03 × 10^−4^
Measured SBP	1.21 (1.01, 1.46)	3.75 × 10^−2^			-	-			1.20 (1.00, 1.44)	0.05			
DBP PRS													
PRS	-	-	0.616	1.36 × 10^−4^	1.52 (1.25, 1.84)	2.16 × 10^−5^	0.619	0	1.50 (1.24, 1.82)	3.34 × 10^−5^	0.630	5.67 × 10^−4^	4.34 × 10^−5^
Measured DBP	1.22 (1.03, 1.44)	1.99 × 10^−2^			-	-			1.20 (1.01, 1.41)	3.28 × 10^−2^			
HTN PRS													
PRS	-	-	0.621	9.64 × 10^−5^	1.19 (0.98, 1.44)	0.08	0.603	0.10	1.17 (0.97, 1.42)	0.11	0.622	1.31 × 10^−4^	0.12
Measured MAP	1.26 (1.06, 1.51)	1.06 × 10^−2^			-	-			1.25 (1.05, 1.50)	1.32 × 10^−2^			
White													
Hypertension													
SBP PRS													
PRS	-	-	0.645	2.56 × 10^−12^	1.29 (1.05, 1.60)	1.60 × 10^−2^	0.614	4.62 × 10^−2^	1.24 (1.00, 1.53)	4.49 × 10^−2^	0.645	9.56 × 10^−12^	0.16
Measured SBP	1.37 (1.23, 1.52)	6.42 × 10^−9^			-	-			1.36 (1.22, 1.51)	1.67 × 10^−8^			
DBP PRS													
PRS	-	-	0.649	1.11 × 10^−15^	1.30 (1.07, 1.59)	8.53 × 10^−3^	0.618	1.03 × 10^−2^	1.29 (1.06, 1.57)	1.23 × 10^−2^	0.651	1.78 × 10^−15^	1.89 × 10^−2^
Measured DBP	1.39 (1.26, 1.55)	2.82 × 10^−10^			-	-			1.39 (1.25, 1.54)	3.89 × 10^−10^			
HTN PRS													
PRS	-	-	0.656	0	1.27 (1.13, 1.44)	1.02 × 10^−4^	0.622	7.00 × 10^−4^	1.25 (1.11, 1.42)	2.54 × 10^−4^	0.661	0	1.79 × 10^−3^
Measured MAP	1.46 (1.31, 1.63)	3.64 × 10^−12^			-	-			1.45 (1.30, 1.61)	8.57 × 10^−12^			
Stage 2 Hypertension													
SBP PRS													
PRS	-	-	0.673	5.71 × 10^−10^	1.33 (1.02, 1.75)	3.84 × 10^−2^	0.648	3.32 × 10^−2^	1.27 (0.97, 1.67)	0.08	0.675	1.57 × 10^−9^	0.10
Measured SBP	1.38 (1.2, 1.58)	3.78 × 10^−6^			-	-			1.37 (1.19, 1.57)	7.33 × 10^−6^			
DBP PRS													
PRS	-	-	0.668	2.59 × 10^−7^	1.42 (1.09, 1.84)	8.57 × 10^−3^	0.652	1.11 × 10^−2^	1.39 (1.07, 1.81)	1.31 × 10^−2^	0.671	4.81 × 10^−7^	2.18 × 10^−2^
Measured DBP	1.34 (1.17, 1.53)	1.77 × 10^−5^			-	-			1.33 (1.16, 1.51)	2.64 × 10^−5^			
HTN PRS													
PRS	-	-	0.675	4.26 × 10^−11^	1.44 (1.22, 1.70)	1.13 × 10^−5^	0.660	1.00 × 10^−4^	1.42 (1.21, 1.67)	2.29 × 10^−5^	0.686	1.26 × 10^−10^	2.24 × 10^−4^
Measured MAP	1.43 (1.24, 1.64)	5.14 × 10^−7^			-	-			1.41 (1.23, 1.62)	1.02 × 10^−6^			

* The null model includes baseline age, baseline age square, sex, baseline BMI, and ancestry principal components. ^§^ Includes variables in the null model plus childhood (baseline) BP. ^†^ Includes variables in the null model plus PRS. ^‡^ Includes variables in the null model plus PRS and childhood (baseline) BP. PRS: Polygenic risk score; BP: Blood pressure; CI: Confidence interval; M1: Model 1; M2: Model 2; M3: Model 3; SBP: Systolic blood pressure; DBP: Diastolic blood pressure; HTN: Hypertension; MAP: Mean arterial pressure.

**Table 3 genes-13-01473-t003:** Associations between PRSs and midlife BP.

	Model 1 ^§^					Model 2 ^†^					Model 3 ^‡^					
BP PRS	Beta	SE	*p*-Value	R^2^	*p*-Value (M1 vs. Null *)	Beta	SE	*p*-Value	R^2^	*p*-Value (M2 vs. Null *)	Beta	SE	*p*-Value	R^2^	*p*-Value (M3 vs. M2)	*p*-Value (M3 vs. M1)
Black (N = 369)																
Systolic blood pressure																
SBP PRS																
PRS	-	-	-	0.07	0.11	5.22	2.45	3.35 × 10^-2^	0.08	3.35 × 10^-2^	4.83	2.46	0.05	0.08	0.17	0.05
Measured SBP	2.29	1.42	0.11			-	-	-			1.96	1.42	0.17			
DBP PRS																
PRS	-	-	-	0.07	0.14	3.45	1.54	2.57 × 10^-2^	0.08	2.57 × 10^-2^	3.24	1.55	3.77 × 10^-2^	0.08	0.21	3.77 × 10^-2^
Measured DBP	2.00	1.34	0.14			-	-	-			1.68	1.35	0.21			
HTN PRS																
PRS	-	-	-	0.07	0.08	1.11	1.59	0.48	0.07	0.48	0.73	1.60	0.65	0.07	0.10	0.65
Measured MAP	2.48	1.43	0.08			-	-	-			2.39	1.44	0.10			
Diastolic blood pressure																
SBP PRS																
PRS	-	-	-	0.06	0.82	2.53	1.49	0.09	0.06	0.09	2.53	1.50	0.09	0.06	0.98	0.09
Measured SBP	0.20	0.87	0.82			-	-	-			0.03	0.87	0.98			
DBP PRS																
PRS	-	-	-	0.07	1.81 × 10^-2^	2.51	0.93	7.45 × 10^-3^	0.07	7.45 × 10^-3^	2.29	0.94	1.48 × 10^-2^	0.09	3.66 × 10^-2^	1.48 × 10^-2^
Measured DBP	1.93	0.81	0.02			-	-	-			1.70	0.81	3.66 × 10^-2^			
HTN PRS																
PRS	-	-	-	0.06	0.07	0.61	0.96	0.52	0.06	0.52	0.37	0.97	0.70	0.06	0.09	0.70
Measured MAP	1.56	0.87	0.07			-	-	-			1.51	0.88	0.09			
White (N = 832)																
Systolic blood pressure																
SBP PRS																
PRS	-	-	-	0.17	1.28 × 10^-9^	4.50	1.31	6.42 × 10^-4^	0.14	6.42 × 10^-4^	3.93	1.29	2.42 × 10^-3^	0.18	4.57 × 10^-9^	2.42 × 10^-3^
Measured SBP	4.04	0.66	1.28 × 10^-9^			-	-	-			3.89	0.66	4.57 × 10^-9^			
DBP PRS																
PRS	-	-	-	0.15	3.81 × 10^-6^	3.58	1.24	3.99 × 10^-3^	0.14	3.99 × 10^-3^	3.41	1.23	5.61 × 10^-3^	0.16	5.30 × 10^-6^	5.61 × 10^-3^
Measured DBP	2.87	0.62	3.81 × 10^-6^			-	-	-			2.82	0.62	5.30 × 10^-6^			
HTN PRS																
PRS	-	-	-	0.17	4.16 × 10^-9^	2.78	0.76	2.60 × 10^-4^	0.14	2.60 × 10^-4^	2.56	0.74	6.27 × 10^-4^	0.18	9.79 × 10^-9^	6.27 × 10^-4^
Measured MAP	3.91	0.66	4.16 × 10^-9^			-	-	-			3.79	0.65	9.79 × 10^-9^			
Diastolic blood pressure																
SBP PRS																
PRS	-	-	-	0.12	3.05 × 10^-5^	2.76	0.90	2.35 × 10^-3^	0.11	2.35 × 10^-3^	2.49	0.90	5.78 × 10^-3^	0.13	7.27 × 10^-5^	5.78 × 10^-3^
Measured SBP	1.92	0.46	3.05 × 10^-5^			-	-	-			1.82	0.46	7.27 × 10^-5^			
DBP PRS																
PRS	-	-	-	0.14	3.17 × 10^-8^	2.73	0.85	1.37 × 10^-3^	0.11	1.37 × 10^-3^	2.59	0.84	2.01 × 10^-3^	0.15	4.62 × 10^-8^	2.01 × 10^-3^
Measured DBP	2.36	0.42	3.17 × 10^-8^			-	-	-			2.32	0.42	4.62 × 10^-8^			
HTN PRS																
PRS	-	-	-	0.14	8.47 × 10^-9^	2.81	0.52	7.00 × 10^-8^	0.13	7.00 × 10^-8^	2.66	0.51	2.00 × 10^-7^	0.17	2.41 × 10^-8^	2.00 × 10^-7^
Measured MAP	2.63	0.45	8.47 × 10^-9^			-	-	-			2.51	0.45	2.41 × 10^-8^			

* The null model includes baseline age, baseline age square, sex, baseline BMI, and ancestry principal components. ^§^ Includes variables in the null plus childhood (baseline) BP. ^†^ Includes variables in the null model plus PRS. ^‡^ Includes variables in the null model plus PRS and childhood (baseline) BP. PRS: Polygenic risk score; BP: Blood pressure; SE: Standard error; M1: Model 1; M2: Model 2; M3: Model 3; SBP: Systolic blood pressure; DBP: Diastolic blood pressure; HTN: Hypertension; MAP: Mean arterial pressure.

## Data Availability

The data that support the findings of this study are available from the corresponding author upon reasonable request.

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
