# Peer review of "Life-Course Associations between Blood Pressure-Related Polygenic Risk Scores and Hypertension in the Bogalusa Heart Study"

_genes, 2022, doi:10.3390/genes13081473_

Round 1

Reviewer 1 Report

The study entitled ‘‘ Life course Associations between Blood Pressure-related Polygenic Risk Scores and Hypertension in the Bogalusa Heart Study ’’ highlights the genetic information utility in the identification the cardiovascular risk subpopulations.

In my opinion, it is well-conducted research, with relevant results, extended over a long period.

I would recommend to the authors:

-             Please explain all the abbreviations used in table 1 (e.g. NA, BMI or SD). The same observation is for table 2.

-             It is sometimes difficult to understand the results presented, especially in table 2. Kindly reconsider the table 2 design by adding rows-lines.

Reviewer 2 Report

Sun et al. investigated the association of PRSs and SBP, DBP, and hypertension in Black and White Bogalusa Heart Study participants recruited in childhood and followed up for 37 years. They concluded that BP PRSs are useful genomic information in early life prediction of hypertension. I have only some minor questions:

1. Why was the BP measured differently in children (i.e, twice with a mercury sphygmomanometer) and adults (i.e., three times with a digital device)?

2. Is there any sex-based differences in the association of PRSs and BP? Please discuss it.
